# Patterns of Focal Amyloid Deposition Using ^18^F-Florbetaben PET in Patients with Cognitive Impairment

**DOI:** 10.3390/diagnostics12061357

**Published:** 2022-05-31

**Authors:** Sung-eun Chung, Hyung-Ji Kim, Sungyang Jo, Sunju Lee, Yoojin Lee, Jee Hoon Roh, Jae-Hong Lee

**Affiliations:** 1Department of Neurology, Asan Medical Center, University of Ulsan College of Medicine, Seoul 05505, Korea; amariju@hanmail.net (S.-e.C.); garailsikzip@gmail.com (H.-J.K.); herbjsy@daum.net (S.J.); microhouse1204@gmail.com (S.L.); san012@naver.com (Y.L.); 2Department of Neurology, Uijeongbu Eulji Medical Center, Eulji University School of Medicine, Uijeongbu 11759, Korea; 3Neuroscience Institute, Korea University College of Medicine, Seoul 02852, Korea; alzheimer@naver.com

**Keywords:** Alzheimer’s disease, amyloid beta 42, conditional probability

## Abstract

Accumulation of aggregated amyloid-β (Aβ) in the brain is considered the first pathological event within the pathogenesis of Alzheimer’s disease (AD). It is difficult to accurately identify the initial brain regions of Aβ accumulation due to the time-lag between the start of the pathophysiology and symptom onset. However, focal regional amyloid uptake on amyloid PET scans may provide insights into this. Hence, we aimed to evaluate the topographic distribution of amyloid deposition in patients with cognitive impairment and to identify the starting order of amyloid accumulation in the brain using conditional probability. We enrolled 58 patients composed of 9 normal cognition (NC), 32 mild cognitive impairment (MCI), and 17 dementia showing focal regional amyloid deposition corresponding to a brain amyloid plaque load (BAPL) score of 2 among those who visited the Memory Clinic of Asan Medical Center and underwent an ^18^F-florbetaben PET scan (March 2013 to April 2019). Regions of interest (ROI) included the frontal, parietal, lateral temporal, and occipital cortices, the posterior cingulate/precuneus, and the striatum. The most frequent occurrence of Aβ deposition was in the posterior cingulate/precuneus (n = 41, 68.3%). The second most frequent site was the lateral temporal cortex (n = 24, 40.0%), followed by the lateral parietal cortex (n = 21, 35.6%) and other lesions, such as the frontal and occipital cortices. The striatum was the least frequently affected. Our study found that the posterior cingulate/precuneus and the lateral temporal and parietal cortices may be the earliest areas to be affected by Aβ accumulation. Longitudinal follow-up of focal brain amyloid deposition may help elucidate the evolutionary pattern of Aβ accumulation in the brain of people with AD continuum.

## 1. Introduction

Amyloid-β (Aβ) and hyperphosphorylated tau deposition are two major pathologic components of Alzheimer’s disease (AD) pathogenesis [1,2]. The accumulation of amyloid plaques in the brain, which occurs for decades prior to the onset of cognitive decline, is the first pathological event within the pathogenesis of AD [3,4]. Until now, patterns of Aβ deposition and propagation have been inferred from a series of postmortem studies [5,6]. However, the development of amyloid positron emission tomography (PET) imaging has made it possible to detect the deposition and spreading of Aβ in vivo [7]. Moreover, amyloid imaging studies are useful not only for the early diagnosis of AD, but also for gaining a better understanding of the pathophysiological events underlying AD [8,9]. Without knowledge of the temporospatial pattern of Aβ accumulation in patients with sporadic AD, there are inherent limitations in understanding the trajectories of amyloid deposition and propagation [10,11,12].

[^18^F]-florbetaben is an ^18^F-labeled polyethylene glycol stilbene derivative with high in vitro affinity and specificity for β-amyloid plaques [13]. Unlike other tracers that are read by dichotomy, the reading of [^18^F]-florbetaben PET is conducted through a triple scale system, termed β-amyloid plaque loading (BAPL) [14,15]. The BAPL scoring system ranges from 1 to 3 according to the amount of regional cortical tracer uptake (RCTU). A BAPL score of 2 (BAPL2) is generally considered an early stage of amyloid accumulation because this stage shows only small amounts and focal deposition of Aβ. In comparing the results of in vivo PET imaging with post-mortem histopathology, [^18^F]-florbetaben was found to exhibit high sensitivity and specificity for pathologically confirmed amyloid plaque detection (97.9% and 88.9%, respectively); moreover, these studies detected a marked and statistically significant association between the results of visual-based qualitative assessments and quantitative analyses [16,17,18,19]. 

There is still some debate as to whether BAPL2 is on the continuum of Alzheimer’s disease or is instead a sign of focal amyloid deposition found within another neurodegenerative disorder. A recent study of patients who underwent a [^18^F]-flutemetamol amyloid PET study reported that focal amyloid deposition refers to a transitional state (i.e., from no amyloid to overt amyloid accumulation) [20]. In this regard, we were interested to analyze the pattern of focal amyloid deposition within BAPL2 in order to investigate its association with the full blown amyloid deposition found in Alzheimer’s dementia. Previous AD studies have shown the conditional probabilities (CP) of successfully predicting the spreading patterns of TDP-43 (TAR DNA binding protein 43) and tau proteins [21,22]. Therefore, in this study, we aimed to explore the early patterns of Aβ deposition using conditional probability calculations.

## 2. Methods

### 2.1. Participants

We conducted a cross-sectional retrospective study, which was performed at a single medical center. More specifically, we reviewed the medical records of patients with memory impairment who visited the Memory Clinic of Asan Medical Center (Seoul, Korea) from March 2013 to August 2019. All patients underwent high-resolution 3.0-Tesla brain magnetic resonance imaging (MRI), primarily to exclude structural lesions (such as from acute stroke, brain tumors, hydrocephalus, and traumatic brain injury) that might influence cognitive function. Cognitive function was assessed through clinical interviews and detailed neuropsychological testing. Blood tests, including a complete blood count, a chemistry battery, evaluations of vitamin B12 and folate levels, syphilis testing, HIV serology, and thyroid function tests were performed in order to rule out the possibility of medical conditions causing cognitive decline. All patients also underwent [^18^F]-florbetaben amyloid PET scans as well as ApoE (apolipoprotein E) genotyping. ApoE genotyping was conducted by extracting genomic DNA with polymerase chain reaction (PCR) from peripheral venous blood [23]. Patients with an intracranial hemorrhage, subdural hemorrhage, acute cerebral infarction and brain tumor were excluded from the dataset. Patients with previous history of seizure, brain surgery, or a severe medical illness due to metabolic causes were excluded. Other cause of dementia, such as idiopathic Parkinson’s disease, corticobasal syndrome, diffuse Lewy body dementia, idiopathic normal hydrocephalus, and frontotemporal degeneration were excluded. Finally, patients who could not conduct detailed neuropsychological test were excluded from the dataset. Ultimately, after applying the study exclusion criteria, a total of 58 patients were included in our dataset.

### 2.2. Neuropsychological Assessments

All participants except two were evaluated using the Seoul Neuropsychological Screening Battery (SNSB) as a standardized neuropsychological assessment [24,25]. The SNSB is a comprehensive neuropsychological battery that includes various tests measuring attention, language, praxis, calculation, verbal and visual memory, visuospatial function, and frontal/executive function. In this study, items in the battery that are convertible to numerical value items were selected for the assessment of each cognitive domain. These items included the results of digit span tests (forward and backward), the Korean version of the Boston Naming Test, the Rey-Osterrieth Complex Figure Test (RCFT; this evaluation includes immediate copying, 20-min delayed recall, and recognition tests), the Seoul Verbal Learning Test (SVLT; this evaluation includes three learning-free recall trials of 12 words, a 20-min delayed recall trial for these 12 items, and a recognition test), a phonemic and semantic Controlled Oral Word Association Test, and a Stroop Test (which includes word and color readings of 112 items during a two minute period). Each score was converted to a standardized score (Z-score) based on age-, sex, and education-adjusted norms. Mini-Mental State Examination (MMSE) and Clinical Dementia Rating sum of boxes scores were used to assess the participants’ global cognitive function. Diagnoses of amnestic MCI (mild cognitive impairment; i.e., prodromal AD) and probable AD were determined by two neurologists (S.J. and S.L). A diagnosis of MCI was determined by applying Peterson’s criteria and a diagnosis of dementia was determined by applying DSM-IV (Diagnostic and Statistical Manual of Mental Disorders) criteria [25,26,27].

### 2.3. [^18^F]-Florbetaben PET Image Acquisition and Analysis

All PET images were acquired using Discovery 690, 710, and 690 Elite PET/CT scanners (GE Healthcare; Milwaukee, WI, USA). [^18^F]-florbetaben amyloid PET images were acquired over the course of 20 min at the dynamic mode (4 × 5 min frames). Scanning was performed 90 to 110 min after injection of a bolus (mean dose, 300 MBq, 8.1 mCi ± 10%) into an antecubital vein.

[^18^F]-florbetaben PET images were assessed by consensus among three physicians (two board-certified staff nuclear medicine physicians at Asan Medical Center and one neurologist [J.H.L]). Interpretation was conducted visually by comparing the activity in the cortical gray matter with activity in the adjacent cortical white matter. In general, only four regions of interest (ROIs; the frontal, temporal, and parietal cortices and the posterior cingulate/precuneus) were interpreted in a visual rating of [^18^F]-florbetaben PET findings. We included the occipital cortex and striatum in these evaluations in order to clearly identify spatial patterns of amyloid deposition. Each of these brain regions was scored according to the RCTU and BAPL systems, described below.

The RCTU scoring system grades the tracer uptake in each of the above regions as follows: 1 = no binding, 2 = minor binding, and 3 = pronounced binding. The RCTU scores are then condensed into a single three-grade scoring system for each PET scan.

BAPL scores are determined as follows: 1 = no β-amyloid load, 2 = a minor β-amyloid load, and 3 = a significant β-amyloid load. BAPL scores of “1” are classified as β-amyloid-negative PET scans, and BAPL scores of “2” and “3” are classified as β-amyloid-positive PET scans [17].

### 2.4. Conditional Probability (CP) Calculations and Statistical Analyses

We were interested in assessing the evidence that one brain region would tend to have earlier β-amyloid involvement than another brain region by evaluating [^18^F]-florbetaben PET images.

To illustrate, we will show a comparison of two regions, X and Y. If there was Aβ in region X, we marked this finding as X+; if there was no Aβ in region Y, we marked this finding as Y− (Figure 1). Conditional probabilities were used to obtain probability values of P (X+|−) and P (Y+|X−), as follows:
$$P\left(Ax+|By-\right)=\frac{n\left(Ax+By-\right)}{\left[n\left(Ax+By-\right)+n\left(Ax-By-\right)\right]}$$

McNemar’s test was used to evaluate the evidence for the null hypothesis that (X+, Y−) and (X−, Y+) occurred equally and that X and Y belong to the same stage. This test was performed in all combinations for six neural regions. CP values thus obtained were displayed on a matrix-like graphical display.

Each cell in the matrix represents a CP, indicating the probability that Aβ will accumulate in the evaluated neural region in preference to other neural regions. Reading the plot from left to right, each item shows the probability that the area on the left is expected to show more positivity than the area on the right. This evaluation likewise displays the probability that the area below the item (reading from top to bottom) is expected to show more positivity than the above area. The theoretical background and details of the CP method for determining the sequential order of the two conditions were described in a previously published theoretical article as well as in a paper that applied this method to evaluations of two proteins [21,22].

All statistical analyses were performed using SPSS statistical software (version 21.0, IBM Corp., Armonk, NY, USA).

## 3. Results

### 3.1. Demographic and Clinical Characteristics

At the time of PET imaging, 29.3% of the enrolled patients had a clinical diagnosis of dementia, 55.2% had a clinical diagnosis of mild cognitive impairment, and 15.5% were evaluated as presenting with cognitive normality (CN). Moreover, 36 (62%) of the enrolled participants were female, the median age at the time of PET imaging was 78.3 years (range: 64 to 92 years) and the median illness duration was 24.7 months (range: 3 to 84 months). The median education level of the enrolled participants was 9.2 years (range: 0 to 22 years) and 21 (36.2%) were ApoE ε4 carriers. When calculating descriptive statistics for vascular risk factors, 29.3% of the enrolled participants were found to have diabetes mellitus (DM), 51.7% had hypertension (HTN), and 37.6% had hyperlipidemia (Table 1).

### 3.2. Neuropsychological Assessment

All patients underwent the SNSB or an equivalent testing scheme and were diagnosed with CN, MCI, or dementia based on the results of these assessments. The detailed neuropsychological test results of 58 patients were displayed in Table 2. The most noticeable difference was observed in the SVLT delayed recall test. As expected, performance worsened in the following order: CN, MCI, and dementia.

### 3.3. CP Analysis

A total of 41 of the 60 (68.3%) evaluated patients had regional amyloid tracer uptake at the posterior cingulate/precuneus, 24 (40%) had deposition in the lateral temporal cortex, and 21 (35%) had deposition in the parietal cortex. Regional amyloid tracer uptake is displayed in Table 3, including with regard to findings in the occipital cortex and the anterior striatum.

CP values were obtained by grouping various combinations of two regions in order to determine the accumulation sequence of region-specific amyloid depositions. Using McNemar’s test, probability values with a *p*-value above 0.05 were deleted. The posterior cingulate/precuneus area, which had the highest frequency of amyloid deposition, was found to be preferentially settled as compared to the other evaluated neural areas, whereas the lateral temporal and parietal cortices were sequentially stacked. Detailed results of the CP analysis are displayed in Figure 2.

We confirmed the statistical significance of the amyloid spreading pattern from the posterior cingulate/precuneus to the lateral temporal cortex, from the posterior cingulate/precuneus to the frontal cortex, from the lateral temporal cortex to the parietal cortex, from the lateral temporal cortex to the anterior striatum, from the parietal cortex to the anterior striatum, and from the parietal cortex to the occipital cortex via CP calculations. The employed calculation method is shown in the Appendix A.

The posterior cingulate/precuneus area was preferentially settled as compared to the other evaluated neural areas. The number in each cell represents the probability of amyloid spreading (higher values indicate higher probabilities of amyloid spreading).

## 4. Discussion

This study evaluated regional amyloid deposition on PET scans in order to investigate the topographic distribution of amyloid deposition as well as to identify the starting order of brain amyloid accumulation in patients with AD continuum. Overall, we observed that the posterior cingulate/precuneus and the lateral temporal and parietal cortices are the areas affected the earliest by Aβ accumulation, and that the striatum is the least affected area.

The CP method described herein proved useful in determining a sequential order of tau and Aβ deposition within a previous study [22]. Although the CP model is not meant to demonstrate causal relationships with regard to protein deposition in the brain, it is appropriate for explaining the precedence of the involvement of one neural area over another based on cross-sectional data [28]. Our results showed that the patterns of amyloid deposition in the BAPL2 patient group, as evaluated using the CP method, were essentially the same as those delineated within Thal’s staging system [29]. Some previous studies had reported that amyloid deposition in the parietal cortex and precuneus did not significantly change along the AD continuum, but recent results reported that regional amyloid deposition could be an important prognostic factor in disease progression [6,11,28].

However, the orbitofrontal cortex is said to be one of the earliest cortical regions to be affected by amyloid deposition based on the results of postmortem examinations [30]. Why this pattern was not apparent in our study remains unanswered. We speculate that amyloid accumulation in the orbitofrontal region may not be optimally assessed by visual readings of PET scans.

To our knowledge, this is the first study to reveal the sequence of amyloid deposition as measured by [^18^F]-florbetaben amyloid PET scanning, though there have been some studies that analyzed amyloid deposition patterns based on post-mortem observations [6,12,30,31]. Our study, however, has the strength of revealing amyloid deposition through analysis of a BAPL2 patient group, a unique condition appearing in [^18^F]-florbetaben amyloid in vivo PET readings [32]. This particular group of patients provide a unique opportunity to look into the topographic pattern of amyloid deposition early in the disease process.

The risk of developing AD is known to increase by approximately 1.8 times in amyloid-positive individuals as compared with amyloid-negative individuals of the same age [33]. With the advent of a new disease modifying therapy aimed at targeting amyloid accumulation, it has become increasingly important to properly select patients in the early stages of amyloid deposition [34]. There is some controversy over whether the BAPL2 amyloid status as determined from florbetaben amyloid PET visual ratings can be considered an eligible condition for the placement on the Alzheimer’s disease continuum. In this study, we showed that amyloid deposition patterns in BAPL2 patients were similar to those seen in AD patients [35]. Amyloid deposition in the occipital cortex and anterior striatum did not appear at a statistically meaningful frequency so as to allow for determining conditional probability. These results support the notion that BAPL2 can be considered an early stage of amyloid deposition in Alzheimer’s patients.

Interestingly, the BAPL2 patients enrolled in our dataset included those with MCI or even those in the dementia (AD) state. According to the amyloid hypothesis, amyloid deposition occurs 15–20 years before dementia symptom onset. As such, if BAPL2 status represents the early stage of amyloid deposition, most BAPL2 patients should be in the asymptomatic or mild symptomatic stage. In this regard, it could be assumed that subthreshold levels of amyloid deposition may be responsible for cognitive impairment in this patient group [36]. Previous studies based on post-mortem observations indicate that subthreshold levels of amyloid plaque implicate low levels of amyloid deposition in the typical regions delineated within the Consortium to Establish a Registry for Alzheimer’s Disease (CERAD) guidelines, and likewise implicate higher levels of amyloid deposition within other specific neural regions [37]. These changes build up a pathological environment similar to that of AD, thus causing cognitive impairment in BAPL2 patients. Moreover, a previous study revealed that subthreshold levels of amyloid deposition could affect subsequent tau deposition [38]. With regard to Braak stage I-II, the baseline Aβ level was found to be the best predictor of consecutive tau deposition, whereas in the progressed stage (Braak III-IV), the accumulation rate of Aβ was found to be the best predictor of subsequent tau deposition. Low levels of amyloid deposition and subsequent tau deposition are considered to be responsible for cognitive symptom presentation in BAPL2 patients [38].

We acknowledge some limitations to this study. For example, the cross-sectional nature of this study is an important limitation. Future longitudinal studies will be necessary in order to better understand how amyloid accumulation occurs and spreads over time. Since the rate of amyloid deposition is an important factor in evaluating the clinical manifestations and prognosis of AD patients, a follow-up study needs to be conducted in order to better understand these associations. Second, tests for other core AD biomarkers, such as tau protein, were not conducted in the current study. This opens the door for the possibility of BAPL2 being non-specific for AD in etiology. However, we tried to exclude other causes of cognitive decline through appropriate diagnostic testing, including with regard to MRI scans, blood laboratory analyses, and other clinical findings. Finally, we evaluated visual rating results rather than quantitative analysis of PET data, such as standardized uptake value ratio (SUVR) values. The concordance rate between cortical SUVR and visual read categorization is known to be excellent and the degree of agreement between visual and quantitative analyses of florebetaben PET in our center is also very high. Moreover, either visual read or quantitative analysis would not matter in determining whether there is regional amyloid uptake in the brain.

## 5. Conclusions

Our study showed that patterns of amyloid deposition in BAPL2 patients analyzed by CP calculations were similar to those found in AD patients. We conclude that BAPL2 amyloid status is more likely to represent the early stage of amyloid deposition of Alzheimer’s disease. However, further longitudinal studies incorporating not only amyloid but also tau imaging findings will be warranted in order to reveal the natural history and the temporospatial spread of amyloid within the BAPL2 patient group.

## Figures and Tables

**Figure 1 diagnostics-12-01357-f001:**
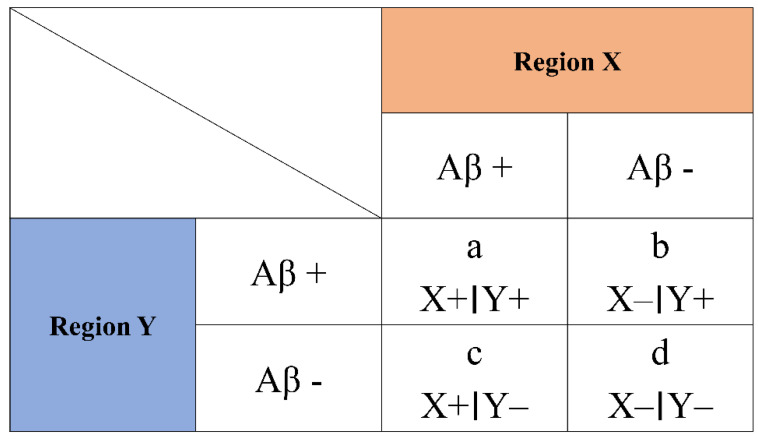
Conceptual scheme of conditional probability (CP) representations. X+ represents positive amyloid deposition in region X, whereas X− indicates that no amyloid deposition was detected in region X. The same matrix was constructed for six regions of interest (ROIs), including the striatum and the occipital cortex.

**Figure 2 diagnostics-12-01357-f002:**
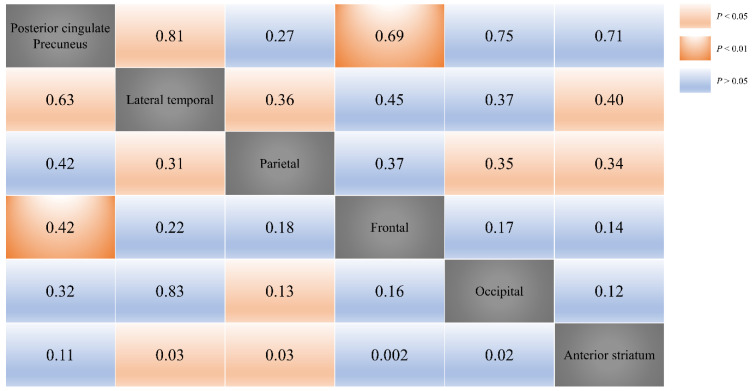
Matrix-like graphical display of results with regard to conditional probability (CP) distributions.

**Table 1 diagnostics-12-01357-t001:** Patient clinical and demographic characteristics.

		CN	MCI	Dementia
Population	Number (%)	9 (15.5)	32 (55.2)	17 (29.3)
	Sex, Female (%)	6 (66.7)	16 (50)	14 (82.3)
	Age (SD) *, y	73.8 (4.5)	77.5 (6.5)	81.9 (6.6)
	Education (SD), y	5.2 (4.9)	10.8 (5.6)	8.8(3.9)
	Symptom duration, (SD), m *	12.2 (7.5)	23.5 (17.8)	30.3 (20.1)
Global cognition	MMSE (SD) * ^+^	25.9 (3.5)	25.3 (4.0)	19.1 (4.9)
	CDR (SD) * ^+^	0.3 (0.3)	0.5 (0.1)	0.7 (0.3)
	CDR-SOB (SD) * ^+^	2.0 (0.7)	3.2 (0.7)	4.2 (0.7)
Vascular risk factors	DM, n (%)	1 (11.1)	10 (31.3)	6 (35.3)
	HTN, n (%)	5 (55.6)	18 (56.3)	7 (41.1)
	Dyslipidemia, n (%)	4 (44.4)	5 (25.0)	7 (41.1)
Genetics	ApoE ε4 carrier, n (%)	3 (33.3)	13 (44.8)	5 (29.4)

Abbreviations: ApoE, Apolipoprotein E; CDR, Clinical Dementia Rating; CDR-SOB, Clinical Dementia Rating Sum of Boxes; DM, Diabetes mellitus; HTN, Hypertension; MMSE, Mini-Mental Status Examination; * Statistically significant at the level of *p* < 0.05; + ANCOVA (analysis of covariance) test with age-adjusted analysis.

**Table 2 diagnostics-12-01357-t002:** Neuropsychologic test results (Seoul Neuropsychological Screening Battery, SNSB).

Domain		CN	MCI	AD
Attention	DS-F	0.4 ± 0.8	0.0 ± 1.0	−0.5 ± 1.2
	DS-B	−0.2 ± 0.9	0.0 ± 1.1	−0.6 ± 1.1
Language	K-BNT *	0.3 ± 0.7	−1.3 ± 1.8	−2.1 ± 2.0
Visuospatial	RCFT copying	−0.8 ± 1.6	−0.4 ± 1.6	−2.0 ± 3.0
Memory	SVLT, immediate *	0.1 ± 0.9	−0.8 ± 0.9	−1.6 ± 1.0
	SVLT, delayed *	0.1 ± 0.9	−1.6 ± 0.9	−2.7 ± 1.6
	SVLT, recognition *	0.4 ± 1.2	−1.2 ± 1.4	−2.2 ± 1.6
	RCFT, immediate *	−0.2 ± 1.0	−1.1 ± 0.8	−1.7 ± 0.5
	RCFT, delayed *	−0.0 ± 0.9	−1.2 ± 0.7	−1.7 ± 0.6
	RCFT, recognition	−0.5 ± 0.9	−0.8 ± 1.0	−1.6 ± 1.7
Frontal/executive	COWAT, Animal *	0.3 ± 1.4	−1.0 ± 1.1	−1.5 ± 1.2
	COWAT, Supermarket *	0.1 ± 1.5	−0.5 ± 0.9	−1.4 ± 0.7
	COWAT, Phonemic *	0.3 ± 1.5	−0.3 ± 1.0	−1.2 ± 0.9
	K-COWAT-CR	−0.7 ± 1.3	−1.2 ± 1.0	−1.7 ± 1.1

Kruskal-Wallis tests were performed to compare differences in neuropsychologic test results among the three evaluated groups. Abbreviations: AD, Alzheimer’s disease; ANCOVA, Analysis of covariance; CN, Cognitive nor mality; COWAT, Controlled Oral Word Association Test; DS, Digit-span test; K-BNT, Korean version Boston Naming Test; MCI, Mild cognitive impairment; RCFT, Rey Complex Figure Test; SVLT, Seoul Verbal Learning Test. * Statistically significant at the level of *p* < 0.05.

**Table 3 diagnostics-12-01357-t003:** Frequencies and distribution with regard to amyloid-β regional uptake.

ROIs	Lateral Temporal	Frontal	Parietal	Posterior Cingulate/ Precuneus	Occipital	Anterior Striatum
Number	24	9	21	41	8	2
Frequency (%)	40	15	35	68.3	13.3	3.3

Abbreviations: ROIs, Regions of interest.

## Data Availability

Due to privacy and ethical concerns, the data that support the findings of this study are available on request from the corresponding authors (J.H.L.).

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
