# Peer review of "Patterns of Focal Amyloid Deposition Using 18F-Florbetaben PET in Patients with Cognitive Impairment"

_diagnostics, 2022, doi:10.3390/diagnostics12061357_

Round 1

Reviewer 1 Report

Chung and colleagues have described a cross sectional study across patients with normal cognition, mild cognitive impairment and dementia had 18F-florbetaben PET followed by inter-regional analysis of amyloid load. They report that 41/60 of the patients have tracer uptake in the posterior cingulate/ precuneus compared to 40% for the next region(lateral temporal cortex) and conclude that this is the former is the site of initial amyloid deposition.

This paper is well-written and easy to follow. However, there are some queries that that the Authors will need to consider (in order of appearance in the manuscript).

1)    Include the number of patients in the Abstract and the breakdown of NC, MCI and Dementia

2)    Page 2, paragraph 1. Prior to comparisons with post-mortem tissue studies, it would be useful to compare the three major amyloid tracers used and why FBB has been chosen here. Also see: https://www.frontiersin.org/articles/10.3389/fnins.2020.00745/full

3)    Page 2, paragraph 3 (and Discussion). Cerebral amyloid angiopathy is more common in the posterior brain, and particularly in APOE4 carriers. How might this affect the conclusions here?

4)    Page 2 Methods – How was the APOE genotyping performed? e.g. PCR of the two SNPs

5)    Methods, last sentence of ‘Participants’. Could the Author check whether it is 58 patients or 60? Also check Table 1 and Line 3 of the Results section (ie) 36/60 = 60% and Page 4, line 3.

6)    Page 3, Paragraph 1, last line – Do you mean DSM-VI?

7)    Page 7, Discussion. Note that Thal et al. reported amyloid the parietal amyloid was seen fewer Phase 1 cases than other lobes (https://n.neurology.org/content/58/12/1791.long. I would also suggest the Authors look at a paper from Peter Nelson’s lab that, similar to Thal et al., suggest a more global distribution of amyloid in the neocortex in early AD and no increase in precuneus per se (https://pubmed.ncbi.nlm.nih.gov/19010392/). The other point the Authors may want to discuss is that the precuneus has been postulated as the region of accelerated amyloid accumulation and hypometabolism in autosomal dominant AD cases (https://www.ncbi.nlm.nih.gov/pubmed/29397305).

Author Response

Re: Response to reviewers’ comments on Manuscript entitled " Patterns of focal amyloid deposition using 18F-florbetaben PET in patients with cognitive impairment".

We are glad that our article was well received by the reviewers and that our article was found to be of interest to your readership. These constructive comments have helped us substantially to improve our work. We further analyzed our data as suggested by the reviews and revised the manuscripts, figures and tables.

Our point-by-point response to the reviewers’ comments is provided below, and revisions to the manuscript are highlighted in yellow in the manuscript. We do hope that you will be satisfied with our clarification on some uncertain points and that the paper may now be formally accepted for publication.

Yours sincerely,

Reviewer 2 Report

Author Chung et al. Focused on regional amyloid deposition on PET scans in order to add a geographic perspective on brain amyloid accumulation in AD patients. This study clarifies an important question about AD research. They are found by beta-amyloid plaque loading (BAPL), measured by 18-F florbetaben PET. The results shown are relevant, but some weaknesses, as a cross-sectional study and a single investigation of the biomarkers of AD reduce the experimental strength. This is a well-written research paper and the length is commensurate with the message.

The literature cited is of important relevance. As a second step, it is suggested to subsequently carry out a longitudinal study, considering this communication as a preliminary investigation. For the above reasons, the article can be accepted for publication without review.

Author Response

Re: Response to reviewers’ comments on Manuscript entitled " Patterns of focal amyloid deposition using 18F-florbetaben PET in patients with cognitive impairment".

We are glad that our article was well received by the reviewers and that our article was found to be of interest to your readership. These constructive comments have helped us substantially to improve our work. We further analyzed our data as suggested by the reviews and revised the manuscripts, figures and tables.

Our point-by-point response to the reviewers’ comments is provided below, and revisions to the manuscript are highlighted in yellow in the manuscript. We do hope that you will be satisfied with our clarification on some uncertain points and that the paper may now be formally accepted for publication.

Yours sincerely,

Author Chung et al. Focused on regional amyloid deposition on PET scans in order to add a geographic perspective on brain amyloid accumulation in AD patients. This study clarifies an important question about AD research. They are found by beta-amyloid plaque loading (BAPL), measured by 18-F florbetaben PET. The results shown are relevant, but some weaknesses, as a cross-sectional study and a single investigation of the biomarkers of AD reduce the experimental strength. This is a well-written research paper and the length is commensurate with the message.

The literature cited is of important relevance. As a second step, it is suggested to subsequently carry out a longitudinal study, considering this communication as a preliminary investigation. For the above reasons, the article can be accepted for publication without review

Answer to reviewer,

 We appreciate your generous comments.

Reviewer 3 Report

The manuscript "Patterns of Focal Amyloid Deposition using 18F-florbetaben PET in patients with Cognitive Impairment" by Chung et al. presents the analysis of Amyloid-beta accumulation patterns in the brains of patients. This analysis allowed the authors to predict the order in which different brain areas start to have PET-detectable deposits of Amyloid-beta peptide. This is a very interesting study, and such prediction is crucial for improving the therapies against Alzheimer's disease. A reliable method to detect the early stages of this disease is desirable because its pathophysiology starts many years before symptoms onset. Although the research presented in the manuscript is cross-sectional retrospective, it is a very good starting point for planning a follow-up longitudinal study. Therefore, the reviewer recommends accepting this manuscript as it is. The only minor issue to correct is panel C in Figure 1, which should read X+|Y- instead of X-|Y+.

Author Response

(The authors gave the same response as above.)
